# Review article: Detection of actionable tweets in crisis events

Anna Kruspe[1], Jens Kersten[2], and Friederike Klan[2]

[1]Technical University of Munich, Munich, Germany
[2]German Aerospace Center (DLR), Jena, Germany

**Correspondence:** Anna Kruspe (anna.kruspe@tum.de)

**Abstract.** Messages on social media can be an important source of information during crisis situations, be they short-term disasters or longer-term events like COVID-19. They can frequently provide details about developments much faster than traditional sources (e.g. official news) and can offer personal perspectives on events, such as opinions or specific needs. In the future, these messages can also serve to assess disaster risks.

One challenge for utilizing social media in crisis situations is the reliable detection of relevant messages in a flood of data. Researchers have started to look into this problem in recent years, beginning with crowd-sourced methods. Lately, approaches have shifted towards an automatic analysis of messages. A major stumbling block here is the question of exactly what messages are considered relevant or informative, as this is dependent on the specific usage scenario and the role of the user in this scenario. In this review article, we present methods for the automatic detection of crisis-related messages (tweets) on Twitter. We start

by showing the varying definitions of importance and relevance relating to disasters, leading into the concept of use case-dependent actionability that has recently become more popular, and is the focal point of the review paper. This is followed by an overview of existing crisis-related social media data sets for evaluation and training purposes. We then compare approaches for solving the detection problem based (1) on filtering by characteristics like keywords and location, (2) on crowdsourcing, and (3) on machine learning technique. We analyze their suitability and limitations of the approaches with regards to actionability.

We then point out particular challenges, such as the linguistic issues concerning social media data. Finally, we suggest future avenues of research, and show connections to related tasks, such as the subsequent semantic classification of tweets.

## 1   Introduction

During a crisis situation, quickly gaining as much information as possible about the tide of events is of crucial importance.

Having access to information is necessary for developing situational awareness, and can mean the difference between life and death. This has become obvious once again in the ongoing COVID-19 pandemic. One source of such information that has started gaining interest in the last couple of years is social media. Twitter users, as an example, write about disaster preparations, developments, recovery, and a host of other topics (Niles et al., 2019). Retrieving this information could lead to significant improvements in disaster management strategies. In contrast to most other information sources, social media posts

show up nearly immediately whenever there is a new occurrence (as long as telecommunication infrastructure is still intact), and as such can deliver information very quickly (Sakaki et al., 2010). Such messages can also provide new perspectives that would not be available any other way at this speed, e.g. ground photos. In addition to factual information, social media can offer personal insights into the occurrences, as well as a back-channel to users for relief providers, government agencies, and other official institutions as well as the media. From a user perspective, 69% of Americans think that emergency response agencies should respond to calls for help sent through social media channels according to a 2010 Red Cross study (American Red Cross, 2010). A very comprehensive overview of social media usage in crisis situations is given in (Reuter and Kaufhold, 2018). Researchers have begun integrating this data source into large cross-national platforms for emergency management, e.g. in the *I-REACT*[1] (Rossi, 2019) and *E2MC*[2] (Havas et al., 2017) projects.

The crux of social media analysis in disasters lies in the reliable retrieval and further analysis, for instance classification, of relevant messages. Twitter users worldwide generate 5,800 tweets per second on average[3]. In any given event, the majority of these posts will not be relevant to the event, or useful to service providers. The question is thus: What messages should be detected during a crisis event, and how can such a detection be implemented? This review article will provide an overview over existing approaches to this problem. We will focus on Twitter data as most other social media sources do not offer a possibility to obtain large amounts of their data to outside researchers, or are not commonly used in a way that facilitates gaining information quickly during a disaster.

In this context, models are commonly trained only once on a fixed set of data, making them inflexible and known to have limited generalization capability in case of new incidents. In contrast, thorough studies conducted by Stieglitz et al. (2018) and Fathi et al. (2020) revealed that interactivity and a customization of social media filtering and analysis algorithms are essential to support responses in various specific crisis situations. In order to take into account this important user-centric perspective, we focus our review not just on pre-trained general-purpose models, but also on adjustable and flexible methods that allow for more interactive data filtering and preparation for further processing.

In the next section, we will examine the problem definition more closely and show why the conventional concepts of "related", "informative", or "relevant" are problematic. Section 3 introduces social media data sets useful for analyzing the task of retrieving tweets of interest, and for training and as testing modeling approaches. In section 4, we will then show how such approaches have been implemented so far, grouped into filtering, crowdsourcing, and machine learning methods. Furthermore, aspects of adapting machine learning methods to the concept of actionability are discussed. Section 5 then goes into detail about the challenges these approaches frequently face, while section 6 briefly describes some related problems. We finish with suggestions for new developments in section 7, and a conclusion in section 8.

---

[1] https://www.i-react.eu/

[2] https://www.e2mc-project.eu/

[3] https://www.omnicoreagency.com/twitter-statistics/

## 2  Problem definition

The task of finding social media posts in a crisis may appear clearly defined at first, but quickly becomes more convoluted when attempting an exact definition. Existing publications have gone about defining their problem statement in a variety of ways. An overview is provided in table 1.

What emerges from this table is a trichotomy between the concepts "related", "relevant", and "informative". Several overlaps between these definitions can be observed. For instance, the class *not related or irrelevant* in (Nguyen et al., 2017a) contains *unrelated* tweets (like in (Burel and Alani, 2018)), but also *related but irrelevant* ones (like class *personal* in (Imran et al., 2013)). Compared to rather subjective classes, like *informative*, *personal* or *useful*, the relatedness to an event is a more objective criterion. As a tentative definition, we subsume that "related" encompasses all messages that make implicit or explicit mention of the event in question. The "relevant" concept is a subset of the "related" concept, comprised of messages that contain actual information pertaining to the event. "Informative" messages, finally, offer information useful to the user of the system, and can be seen as a subset of "relevant" in turn. Not all publications necessarily follow this pattern, and lines between these concepts are blurry. In reality, many border cases arise, such as jokes, sarcasm, and speculation. In addition, the question of what makes a tweet informative, or even relevant, is highly dependent on who is asking this question, i.e. who the user of this system is. Such users are often assumed to be relief providers, but could also be members of the government, the media, affected citizens, their family members, and many others. Building on top of this, each of these users may be interested in a different use case of the system, and the employed categorization may be too coarse for their purposes. For instance, humanitarian and governmental emergency management organizations are interested in understanding "the big picture", whereas local police forces and firefighters desire to find "implicit and explicit requests related to emergency needs that should be fulfilled or serviced as soon as possible" (Imran et al., 2018). These requirements also strongly depend on the availability of information from other sources, e.g. government agencies or news outlets.

In recent years, researchers have begun to address these challenges by introducing the concept of "actionability" to describe information relevance from the end user perspective of emergency responders (He et al., 2017) as opposed to generalized situational awareness. Zade et al. (2018) loosely define actionability as "information containing a request or a suggestion that a person should act on and an assumption that a message actionable to some responders may be irrelevant to others", while McCreadie et al. (2020) specify it implicitly via certain topical classes. The concept "serviceability" as introduced in (Purohit et al., 2018) is similar, but only applies to messages directly addressed to relief providers and is defined more narrowly. Similarly, according to (Kropczynski et al., 2018), a "golden tweet" – a post on Twitter containing actionable information for emergency dispatch and supporting the immediate situational awareness needs of first responders – should contain information that addresses the well known five W's (where, what, when, who, why) as well as information on weapons.

In this paper, we define an actionable tweet in a crisis event as one that is relevant and informative in a certain use case or to a certain user. Naturally, focusing on user-centric actionability adds complexity to the corresponding methodological and technical systems, and many of the presented methods do not yet offer this flexibility. However, we believe that this is a viable path forward to make such systems more useful in real-life situations. For the remainder of the paper, we will point out how

**Table 1.** Overview of class definitions for filtering crisis-related tweets

| Article | Class | Definition |
| --- | --- | --- |
| (Imran et al., 2013) | Personal | A message only of interest to its author and her immediate circle of family/friends - does not convey any useful information to people who do not know its author |
| | Informative | Messages of interest to other people beyond the author's immediate circle |
| | Other | Not related to the disaster |
| (Parilla-Ferrer et al., 2014) | Informative | A tweet provides useful information to the public and is relevant to the event |
| | Uninformative | Tweets that are not relevant to the disaster and these do not convey enough information or are personal in nature and may only be beneficial to the family or friends of the sender |
| (Caragea et al., 2016) | Informative | Useful information |
| | Not informative | Not relevant to the event and no useful information |
| (Win and Aung, 2017) | Informative | Useful information |
| | Not informative | Not relevant to the event and no useful information |
| | Other information | Messages related to the event but without useful information |
| (Nguyen et al., 2017a) | Useful/Relevant | Information that is useful to others |
| | Not related or irrelevant | Not related to the event or does not contain useful information for others |
| (Burel and Alani, 2018) | Crisis related | Message related to a crisis situation in general without taking into account informativeness or usefulness |
| | Non-crisis related | Message that is not related to a crisis situation |
| (Stowe et al., 2018) | Relevant | Any information that is relevant to disaster events, including useful information but also jokes, retweets, and speculation |
| | Irrelevant | Not related to a disaster event |

existing data sets and methods can be enhanced in the future to make systems adaptable to individual requirements by different users. We deliberately do not focus on specific use cases, but rather on approaches to guarantee this adaptability.

An aspect that is often neglected in social media-based crisis analytics is the existence of mature and well-established workflows for disaster response activities that have so far been mainly based on geo-data and remote sensing (Voigt et al., 2016; Lang et al., 2020). Information from social media channels should therefore not been seen as solitary but rather as an additional, complementary source of information. In this context, further interesting use-cases, corresponding questions and problem definitions arise in which social media may fill temporal gaps between satellite data acquisitions, could be used to identify

areas that need assistance, and to trigger local surveys.

## 3   Data sets

Collections of social media data created during crises are necessary to study what users write about, how this develops over time, and to create models for automatic detection and other tasks. For these reasons, several such data sets have already been created. As mentioned above, Twitter is the most salient source of data for this use case; therefore, available data sets are mainly composed of Twitter data.

Table 2 lists an overview of available Twitter data sets collected during disaster events. These mainly focus on the text content of tweets, except for *CrisisMMD* which provides tweets with both text and images. Some of these data sets only contain data for one event, while others aggregate multiple ones. Based on various existing data sets, Wiegmann et al. (2020a) recently proposed a balanced compilation of labeled Tweets from 48 different events covering the ten most common disaster types. A distinction can also be made for corpora focusing on natural disasters and those also including man-made disasters. *Events2012* goes even further, containing around 500 events of all types, including disasters.

Annotations vary between these data sets. Some of them do not contain any labels beyond the type of event itself, while others are labeled according to content type (e.g. "Search and rescue" or "Donations"), information source (first-party observers, media, etc.), and priority or importance of each tweet (*CrisisLexT26* and *TREC-IS 2019B*).

A general issue with these data sets lies in the fact that researchers cannot release the full tweet content due to Twitter's redistribution policy[4]. Instead, these data sets are usually provided as lists of tweet ID's, which must then be expanded to the full information ("hydrated"). This frequently leads to data sets becoming smaller over time as users may choose to delete their tweets or make them private. For instance, as of September 2020, only ~30 % of all labeled Tweets from the *Events2012* data set are available. Additionally, the teams creating these corpora have mainly focused on English- and occasionally Spanish-language tweets to facilitate their wider usage for study. More insights would be possible if tweets in the language(s) of the affected area were available. However, Twitter usage also varies across countries. Another factor here is that less than 1% of all tweets contain geolocations (Sloan et al., 2013), which are often necessary for analysis. The following sections provide descriptions of the data sets in more detail:

**Events2012**   This data set was acquired between October 9 and November 7 in 2012 and contains 120 million tweets, of which around 150,000 were labeled to belong to one of 506 events (which are not necessarily disaster events) (McMinn et al., 2013). The event types are categorized into eight groups, such as "Business & Economic" "Arts, Culture & Entertainment", "Disasters & Accidents", or "Sports".

**CrisisLexT6 and T26**   *CrisisLexT6* was first published by Olteanu et al. (2014) and expanded later to *CrisisLexT26* (Olteanu et al., 2015). The sets contain tweets collected during 6 and 26 crises, respectively, mainly natural disasters like earthquakes, wildfires and floods, but also human-induced disasters like shootings and a train crash. Amounts of these tweets per disaster range between 1,100 and 157,500. In total, around 285,000 tweets were collected. They were then annotated

---

[4]https://developer.twitter.com/en/developer-terms/agreement-and-policy

**Table 2.** Overview of crisis-related Twitter data sets

| Name | # Labeled tweets | # Total tweets | Labeled concepts (#classes) | Covered event types |
|---|---|---|---|---|
| **Events2012** (McMinn et al., 2013) | ~150,000 | 120 mio. | 506 Events (8) | Disasters and accidents, other events in sports, arts, culture and entertainment |
| **CrisisLexT6** (Olteanu et al., 2014) | ~6,000 | ~6,000 | Relatedness (2) | Hurricane, flood, bombing, tornado, explosion |
| **CrisisLexT26** (Olteanu et al., 2015) | 26,000 | 285,000 | Informativeness (2), information type (6), tweet source (6) | Earthquake, flood, wildfire, meteor, typhoon, flood, explosion, bombing, train crash, building collapse |
| **Disasters on Social Media (DSM)** (Crowdflower, 2015) | ~10,000 | ~10,000 | Relevance (4) | Not provided |
| **Incident-related Twitter Data (IRTD)** (Schulz and Guckelsberger, 2016) | ~21,000 | ~21,000 | Relatedness (2), incident type (4) | Crash, fire, shooting |
| **CrisisNLP** (Imran et al., 2016b) | 23,000 | 53 mio. | Information type (9) | Earthquake, hurricane, flood, typhoon, cyclone, ebola, MERS |
| **CrisisMMD** (Alam et al., 2018b) | 16,000 | 16,000 | Informativeness (2), information type (8), 3 damage severity (3) | Hurricane, earthquake, wildfire, flood |
| **Epic** (Stowe et al., 2018) | ~25,000 | 25,000 | Relevance (2), information type (17), sentiment (3) | Hurricane |
| **Disaster Tweet Corpus 2020 (DTC)** (Wiegmann et al., 2020b, a) | ~150,000 | ~5.1 mio. | Relatedness (2) | Biological, earthquake, tornado, hurricane, flood, industrial, societal, transportation, wildfire |
| **TREC-IS 2019B** (McCreadie et al., 2019, 2020) | ~38,000 | ~45,000 | Information type (25), priority (4), actionability (2) | Bombing, earthquake, flood, typhoon/hurricane, wildfire, shooting |
| **Appen Disaster Response Messages** (Appen Ltd., 2020) | ~30,000 | ~30,000 | Information type (36) | Earthquake, flood, hurricane |
| **Storm-related Social Media (SSM)** (Grace, 2020) | ~22,000 | 22,000 | Relatedness (2), information type (19), aggregated information type (6), 3 toponym concepts (2/2/3) | Tornado |

by paid workers on the *CrowdFlower* crowdsourcing platform[5] according to three concepts: Informativeness, information type, and tweet source.

**Disasters on Social Media (DSM)** This resource is available on *CrowdFlower*[6] and contains around 10,000 tweets that were identified via keyword-based filtering (for example "ablaze", "quarantine", and "pandemonium"). At its finest granularity, four different classes are distinguished: (1) Relevant (65.52 %), (2) Not Relevant (27.59 %), (3) Relevant Can't Decide (4.6 %), and (4) Not Relevant Can't Decide (2.3 %). No information regarding the covered event types is available, but a cursory review of the data reveals that a multitude of events is found with the keywords, e.g. floods, (wild)fires, car crashes, earthquakes, typhoons, heat waves, plane crashes, terrorist attacks, etc.

**Incident-related Twitter Data (IRTD)** Within three time periods in 2012–2014, around 15 million tweets in a 15 km radius around the city centers of Boston (USA), Brisbane (AUS), Chicago (USA), Dublin (IRE), London (UK), Memphis (USA), New York City (USA), San Francisco (USA), Seattle (USA) and Sidney (AUS), were collected. After filtering by means of incident-related keywords, redundant tweets and missing textual content, the remaining set of around ~21,000 tweets was manually labeled by five annotators using the *CrowdFlower* platform. The annotators labeled according to two different concepts: (1) 2 classes: "incident related" and "not incident related", and (2) 4 classes: "crash", "fire", "shooting", and a neutral class "not incident related". Manual labels for which the annotator agreement was below 75 % were discarded (Schulz and Guckelsberger, 2016).

**CrisisNLP** The team behind *CrisisNLP* collected tweets during 19 natural and health-related disasters between 2013 and 2015 on the *AIDR* platform (see section 4.2) using different strategies (Imran et al., 2016b). Collected tweets range between 17,000 and 28 million per event, making up around 53 million in total. Out of these, around 50,000 were annotated both by volunteers and by paid workers on *CrowdFlower* with regard to nine information types.

**CrisisMMD** *CrisisMMD* is an interesting special case because it only contains tweets with both text and image content. 16,000 tweets were collected for seven events that took place in 2017 in five countries. Annotation was performed by *Figure Eight* for text and images separately. The three annotated concepts are: Informative/Non-informative, eight semantic categories (like "Rescue and volunteering" or "Affected individuals"), and damage severity (only applied to images) (Alam et al., 2018b).

**Epic** This data set with a focus on Hurricane Sandy was collected in a somewhat different manner than most others. The team first assembled tweets containing hashtags associated with the hurricane, and then aggregated them by user. Out of these users, they selected those who had geotagged tweets in the area of impact, suggesting that these users would have been affected by the hurricane. Then, 105 of these users were selected randomly, and their tweets from a week before landfall to a week after were assembled. This leads to a data set that in all probability contains both related and unrelated

---

[5]Later named *Figure Eight*, https://www.figure-eight.com/; acquired in 2019 by *Appen*, https://appen.com
[6]https://data.world/crowdflower/disasters-on-social-media

tweets by the same users. Tweets were annotated according to their relevance as well as 17 semantic categories (such as "Seeking info" or "Planning") and sentiment (Stowe et al., 2018).

**Disaster Tweet Corpus 2020 (DTC)** This data set contains tweets collected, annotated, and published in several other works (Imran et al., 2014; Olteanu et al., 2014, 2015; Imran et al., 2016c; Alam et al., 2018b; Stowe et al., 2018; McMinn et al., 2013), and covers 48 disasters over 10 common disaster types. This balanced collection is intended as a benchmarking data set for filtering algorithms in general (Wiegmann et al., 2020b, a). Additionally, a set of 5 million unrelated tweets, collected during a tranquil period, i.e., where no disasters happened, is provided. This is intended to test filtering models in terms of false positive rates.

**TREC-IS 2019B** A crisis classification task named "Incident Streams" has been a part of the Text REtrieval Conference (TREC) organized by NIST since 2018 (McCreadie et al., 2019). In the first iteration, tweets for six events were first collected automatically using a pre-defined list of keywords, and then annotated with one of 25 information type categories. Further iterations were conducted twice in 2019, for which the data set was expanded each time through a sophisticated process of crawling Twitter and then downsampling the results. The format was also changed to allow multiple labels per tweet. There are several subsets that have been flexibly used for training and testing in the task, partially comprised of *CrisisNLP* and *CrisisLex*. We show the 2019B iteration here, but each iteration has been composed of somewhat different data, comprising 48 crisis events, 50,000 tweets, and 125,000 labels in total. In the 2020 iterations, only events that took place in 2019 were included (McCreadie et al., 2020). *TREC-IS* also contains a concept of actionability defined by a selection of the semantic classes.

**Appen Disaster Response Messages** This data set was published in an open-source format originally by *Figure Eight*, now part of private company *Appen* (Appen Ltd., 2020). It contains 30,000 messages split into training, test, and validation sets collected during various disaster events between 2010 and 2012. These tweets are annotated according to 36 content categories, such as "Search and rescue", "Medical help", or "Military", as well as with a "Related" flag. These messages contain multiple languages plus English translations. The data set also includes news articles related to disasters. The data set is used in a Udacity course[7] as well as a Kaggle challenge[8].

**Storm-related Social Media (SSM)** Presented in (Grace, 2020), this data set was collected during a 2017 tornado in Pennsylvania using three methods: Filtering by Twitter-provided geolocation in the affected area; keyword filtering by place names in the affected area; and filtering by networks of users located in the affected county. For the last approach, user IDs are available in a supplementary data set. Tweets were then labeled according to six concepts: Relatedness to the storm; semantic information type (subsumed from other publications, e.g. (Olteanu et al., 2015)); an aggregated set of the semantic information types (e.g. disruptions, experiences, forecasts); and three toponym-related concepts. Labeling

---

[7]https://www.udacity.com/course/data-scientist-nanodegree--nd025
[8]https://www.kaggle.com/jannesklaas/disasters-on-social-media

was done by three assessors for part of the data set, then split between them for the rest, after consolidating discrepancies. The data is available as supplementary material for (Grace, 2020)[9].

All presented data sets offer advantages and disadvantages, depending on the use case. Almost all of them contain information type annotations, but there is no universal agreement on an ontology here. Many of the used information type definitions are compatible across data sets, but this requires manual work. In addition, interpretation that may lead to errors is required, on the one hand because the classes are often not clearly defined, and on the other because even the meanings of classes with the same name can vary between data sets. The information type ontology provided in *TREC-IS 2019B* was developed and refined in collaboration with help providers, and could therefore be a valuable basis for future annotations.

In published works, *CrisisNLP* and *CrisisLexT26* are used most frequently to demonstrate novel approaches because they are relatively large and cover a wide range of event types. As mentioned above, the *Appen* material is used in Udacity courses and on Kaggle, and may therefore also be a useful starting point for new researchers. For detection of disaster-related tweets, *Events2012* is also very interesting because it contains both disaster events as well as other events, and is much larger than the two others. It does not contain information type annotations, however.

All four of these data sets contain tweets created before 2017, which is relevant because the character limit for tweets was increased from 140 to 280 in 2017. For a large data set of newer tweets, the latest iteration of the *TREC-IS* set is very interesting. In addition, existing approaches for this data set can be recreated from the TREC challenge. *CrisisMMD* has not been used as frequently so far, but is interesting because of the added image content. This data set as well as *Epic* and *SSM* does not cover as many different events, but in exchange, they have a much wider selection of labeled concepts that have not received as much attention so far. *DTC* is interesting due to its aggregation of several data sets and resulting large size and wide coverage, making it usable for benchmarks.

All of these data sets operate under the notion of "related"/"informative"/"relevant" tweets, either by providing explicit labels for these concepts, or by assuming that all contained tweets belong to these concepts. As described in section 2, these conventional annotations are too rigid to implement a detection of actionability for different use cases. We suggest two solutions for future systems:

1. Explicitly annotating tweets with use case-dependent actionability labels. This is, of course, a costly option, but would be highly interesting as a starting point for developing adaptable systems.

2. Defining actionability in a use case-specific way as a composite of other (basic) concepts. A data set labeled with those basic concepts could then be used for different use cases. This is, for example, done in the *TREC-IS 2019B* data set through a selection of information type classes, primarily request and report classes. With the refined ontologies of information types and other concepts contained in the presented data sets, individual profiles of relevant concepts and event types could be created per use case to define actionability in future research. These profiles could even be inferred by automatic models.

---

[9]https://www.sciencedirect.com/science/article/pii/S2352340920304893

## 4 Approaches

As described above, users generate huge amounts of data on Twitter every second, and finding tweets related to an ongoing event is not trivial (Landwehr and Carley, 2014). Several detection approaches have been presented in literature so far. We will group them into three categories: Filtering by characteristics, crowdsourcing, and machine learning approaches. As researchers have only started to focus on detecting actionable information in recent years, many of the presented methods do not offer the necessary flexibility yet, instead only offering solutions for specific use cases or the generalized task of finding related/relevant/informative tweets in a crisis event. Nevertheless, we will present them here as a very useful basis for future work, and will point out whether the described approaches are already useful for detecting actionable information or how they can be adapted accordingly. These questions are somewhat easier to answer for filtering by characteristics and crowdsourcing (sections 4.1 and 4.2) because such systems need to be invoked for specific tasks in a new event anyway. For machine learning methods however, models are usually trained on data from past events or tasks and then statically used in newly occurring events, as described in section 4.3. In section 4.4, we point out novel directions of research for also adapting machine learning algorithms to desired new tasks, implementing the actionability concept.

### 4.1 Filtering by characteristics

The most obvious strategy is the filtering of tweets by various surface characteristics. An example is *TweetTracker*, which was first presented in 2011 (Kumar et al., 2011) and is still available[10]. This platform is able to collect tweets by hashtag, keyword, or location, perform trend analysis, and provide visualizations.

Keywords and hashtags are used most frequently for this and often serve as a useful pre-filter for data collection (e.g. in (Lorini et al., 2019) where tweets are pre-filtered by geographic keywords). The Twitter API allows searching directly for keywords and hashtags or recording the live stream of tweets containing those, meaning that this approach is often a good starting point for researchers. This is especially relevant because only 1 % of the live stream can be collected for free (also see section 5) - when a keyword filter is employed, this 1 % is more likely to contain relevant tweets.

Olteanu et al. (2014) developed a lexicon called *CrisisLex* for this purpose. However, the keyword-filtering approach easily misses tweets that do not mention the keywords specified in advance, particularly when changes occur or the attention focus shifts during the event. To tackle this recall-related problem, Olteanu et al. (2014) propose a method to update the keyword list based on query expansion using new messages. A further, semi-supervised dynamic keyword generation approach, utilizing incremental clustering, SVMs, expectation maximization and word graph generation, is proposed in (Zheng et al., 2017).

Another problem with keyword lists is that unrelated data that contains the same keywords may be retrieved (Imran et al., 2015). In general, filtering by keywords is not a very flexible approach to tackle different use cases and therefore implement actionability. Nevertheless, such approaches have been used in insightful studies, e.g. in (de Albuquerque et al., 2015), where keyword-filtered tweets during a flood event were correlated with flooding levels.

Geolocation is another frequently employed feature that can be useful for retrieving tweets from an area affected by a disas-

---

[10]http://tweettracker.fulton.asu.edu/

ter. However, this approach misses important information that could be coming from a source outside the area, such as help providers or news sources. Additionally, only a small fraction of tweets is geo-tagged at all, leading to a large amount of missed tweets from the area (Sloan et al., 2013).

## 4.2 Crowdsourcing approaches

To resolve the problems mentioned above, other strategies were developed. One solution lies in crowdsourcing the analysis of tweets, i.e. asking human volunteers to manually label the data (Poblet et al., 2014). From an actionability standpoint, this may seem ideal because human subjects are fairly good judges of whether a tweet is relevant in a specific use case. However, this seemingly easy task can easily turn into a complex problem that is subject to the individual volunteers' interpretation depending on the situation. Partitioning the problem into sub-tasks that can be judged more easily can be a remedy to this (Xu et al., 2020).

The main disadvantage of crowdsourcing lies in the necessity for many helpers due to the large amount of incoming tweets, and the resulting effort necessary to organize tasks and consolidate results. Nevertheless, volunteers can be extremely helpful in crisis situations. Established communities of such volunteers exist and can be activated quickly in a disaster event, for example the *Standby Task Force*[11].

To facilitate their work, platforms have been developed over the years. One of the most well-known systems is *Ushahidi*[12]. This platform allows people to share situational information in various media, e.g. by text message, by e-mail, and of course by Twitter. Messages can then be tagged with categories relevant to the event. *Ushahidi* was started by a team of Kenyan citizens during the 2007 Kenyan election crisis, and has since been used successfully in a number of natural disasters, humanitarian crises, and elections (for monitoring). Both the server and the platform software are available open-source[13]. Efforts were made to integrate automatic analysis tools into the platform (named "SwiftRiver"), but discontinued in 2015.

Such automatic analysis tools are the motivation for *AIDR* (Imran et al., 2015). *AIDR* was first developed as a quick response to the 2013 Pakistan earthquake. Its main purpose lies in facilitating machine learning methods to streamline the annotation process. In a novel situation, users first choose their own keywords and regions to start collecting a stream of tweets. Then, volunteers annotate relevant categories. A supervised classifier is then trained on these given examples, and is automatically applied to new incoming messages. A front-end platform named *MicroMappers*[14] also exists. *AIDR* is available in an open-source format as well[15]. It has been used in the creation of various data sets and experiments.

Another contribution to crowdsourcing crisis tweets is *CrisisTracker* (Rogstadius et al., 2013). In *CrisisTracker*, tweets are also collected in real-time. Local Sensitive Hashing (LSH) is then applied to detect clusters of topics (so-called stories), so that volunteers can consider these stories jointly instead of single tweets. The *AIDR* engine has also been integrated to provide

---

[11]https://www.standbytaskforce.org/
[12]https://www.ushahidi.com/
[13]https://github.com/ushahidi/Ushahidi_Web
[14]https://micromappers.wordpress.com/
[15]https://github.com/qcri-social/AIDR

topic filtering. As a field trial, the platform was used in the 2012 Syrian civil war. *CrisisTracker* is also available free and open-source[16], but maintenance stopped in 2016.

## 4.3 Machine learning approaches

To forgo the need for many human volunteers while still intelligently detecting crisis-related tweets, various machine learning approaches have been developed over the years. We distinguish between two categories here: "Traditional" machine learning approaches that put an emphasis on NLP feature engineering, and deep learning approaches with Neural Networks that often utilize automatically learned word or sentence embeddings. An overview of proposed methods of both types is given in table 3.

Generally, machine learning approaches all follow the same rough processing pipeline which is outlined in figure 1. Pre-processed text data is fed into a feature extraction method, and the generated features are forwarded to a model that then outputs a result. In deep learning approaches, this model is a neural network. Feature extraction and model training/inference used to be separate processes in classical NLP, but have become increasingly combined over the past years with the arrival of word and sentence embeddings that can be integrated into the training process.

In both flavors of machine learning, research has mainly focused on static general-purpose models trained a single time on known data to reduce social media information overload. These models are usually intended to detect messages that are potentially relevant to crisis situations. An immediate applicability comes at the cost of a limited generalization capability, i.e. in case of new events and especially new event types, the models may fail dramatically (see for example experimental results in (Wiegmann et al., 2020b)). Furthermore, a decision is usually made on tweet-level without taking into account thematically, spatially or temporally adjacent information. As pointed out in section 2, it is now becoming apparent that more user-centric perspectives need to be taken into account (i.e. defining actionability for a certain task). Hence, more adjustable and flexible methods that allow for more interactive data filtering by actionability are also reviewed here (see section 4.4). These methods do not necessarily focus on the filtering task itself, but can be used in this context and may provide additional valuable capabilities, like an aggregation of semantically similar messages, to support the understanding of contained information and their changes.

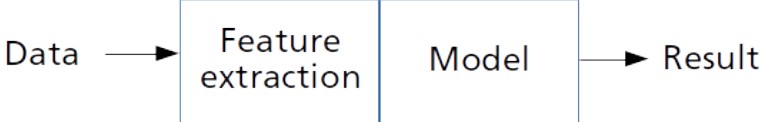

**Figure 1.** General processing pipeline for machine learning approaches.

---

[16]https://github.com/JakobRogstadius/CrisisTracker/

**Table 3.** Overview of the related work proposing filtering algorithms, ordered by the employed method, and listing the data sets used.

| Reference | Features | Method | Data |
|---|---|---|---|
| *Machine learning based on feature engineering* | | | |
| Parilla-Ferrer et al. (2014) | BoW | NB, SVM | Own data |
| Stowe et al. (2016) | Time, retweet, URLs, unigrams, NER, POS, *Word2vec* | NB, Maximum Entropy, SVM | Own data |
| To et al. (2017) | BoW, TF-IDF (with PCA) | LR | *CrisisLexT26, DSM* |
| Win and Aung (2017) | POS, n-grams, emotions, word cluster, lexicon-based features, URLs, hashtags | Linear classification, SVM | *CrisisLexT6* |
| Habdank et al. (2017) | Term counts, TF-IDF, n-grams | NB, Decision Tree, SVM, RF, ANN | Own data |
| Resch et al. (2018) | BoW | LDA | Own data |
| Li et al. (2018) | Term occurrence | NB, semi-supervised domain adaptation | *CrisisLexT6* |
| Mazloom et al. (2019) | BoW | NB, RF, domain adaptation | *CrisisLexT6, IRTD* |
| Kejriwal and Zhou (2019) | *fastText* | Linear Regression ensemble, semi-supervised | Own data |
| Kaufhold et al. (2020) | BoW, TF-IDF, NER, author-event distance, RT, URLs, media, tweet length, language | RF: active, incremental and online learning | Own data |
| *Neural networks* | | | |
| Caragea et al. (2016) | BoW, n-grams | CNN | *CrisisLexT26* |
| Nguyen et al. (2016b) | Domain-specific *Word2vec* | CNN, online training | *CrisisNLP* |
| Nguyen et al. (2017a) | *Word2vec* (Mikolov et al., 2013), own crisis word embedding, | CNN | *CrisisLexT6, CrisisNLP* |
| Alam et al. (2018a) | *Word2vec*, graph embedding | CNN, adversarial and semi-supervised learning | *CrisisNLP* |
| Burel and Alani (2018) | *Word2vec* | CNN | *CrisisLexT26* |
| Kersten et al. (2019) | *Word2vec* (Imran et al., 2016a) | CNN | *CrisisLexT26, CrisisNLP, Epic, Events2012, CrisisMMD* |
| Kruspe et al. (2019) | *Word2vec* (Nguyen et al., 2016a) | CNN few-shot model | *CrisisLexT26, CrisisNLP* |
| Ning et al. (2019) | Autoencoder: Linguistic, emotional, symbolic, NER, LDA | CNN | *CrisisLexT26* |
| Lorini et al. (2019) | Multilanguage-adapted *GloVe* | CNN | Own data (floods) |
| Wiegmann et al. (2020b) | USE, BERT, (Imran et al., 2016a) | CNN, NN | *DTC* |
| Snyder et al. (2020) | *Word2vec* | CNN, RNN, LSTM | *CrisisLexT26*, Own data |
| de Bruijn et al. (2020) | Multilanguage-adapted *fastText* | CNN + multimodal NN | Own data (floods) |

### 4.3.1 Machine learning based on feature engineering

**Linguistic features**

A crucial component of a social media classification model is the representation of the text data at the input (i.e. how words or sentences are mapped to numeric values that the model can process). Classical NLP features are based in linguistics and may employ additional models, e.g. for sentiment analysis or topic modeling.

A corpus (i.e. set) of documents (i.e. tweets) is built up by a vocabulary of $N$ words. A straightforward approach to represent each word is a "one-hot" vector of length $N$. Given the $i^{th}$ word of the vocabulary, the corresponding one-hot vector is 1 at position $i$ and zero otherwise. Depending on the vocabulary size, these vectors might be quite large and the one-hot representation does not allow for direct comparison of different words, e.g. with Euclidean or Cosine similarity.

Within this framework, a Bag-of-Words (BoW) model simply counts the occurrence of each term (term frequency–TF) in a document or corpus independently of its position. In order to reduce the impact of frequently occurring but not descriptive terms, like "a" or "and", these so-called stop words can be removed in advance or the term frequencies are normalized, for example by the commonly used inverse document frequency (IDF). TF-IDF results in high weights in case of a high term frequency (in a document) along with a low term frequency over the whole corpus. Even though this approach proved to be suitable in many studies (Parilla-Ferrer et al., 2014; To et al., 2017; Resch et al., 2018; Mazloom et al., 2019; Kaufhold et al., 2020), contextual information is neglected. The concept of n-grams accounts for context in terms of $n$ adjacent terms. However, this approach may drastically increase the vocabulary dimensionality.

Further commonly used features (see for example (Stowe et al., 2016; Kaufhold et al., 2020)) result from part-of-speech (POS) tagging and named entity recognition (NER). POS tagging finds the syntactic category of each words (e.g., noun, verb or adjective) in written text, whereas NER allows for tagging all words representing given names, for example of countries, places, companies, and persons. The extracted features are sometimes subjected to dimensionality reduction procedures such as Principal Component Analysis (PCA) before the model input.

Finally, Twitter-specific features, like tweet length, timestamp, whether a tweet is a retweet, whether a tweet contains media, links, emojis, usernames, or hashtags, have been found to be useful features (see for example (Stowe et al., 2016; Win and Aung, 2017; Kaufhold et al., 2020)).

A few approaches also use neural network-based word embeddings, e.g. *Word2vec* and *fastText*, which are described below.

**Models**

Based on the feature vectors that represent a tweet, several methods are available to train models that seek to assign each tweet to pre-defined classes. The task of distinguishing crisis- or incident-related content from all other types of tweets is a binary problem, for which generative and discriminative approaches exists. Generative approaches attempt to model the joint probability of the features and the corresponding labels. Even the relatively simple Naïve Bayes approach produces promising results, for example in (Parilla-Ferrer et al., 2014; Stowe et al., 2016; Habdank et al., 2017; Mazloom et al., 2019).

In contrast, discriminative methods, like Support Vector Machines (SVMs), decision trees, Random Forests (RFs) and Logistic

Regression (LR), are commonly used to directly distinguish between classes (see for example (Win and Aung, 2017; Kejriwal and Zhou, 2019)). For instance, a linear SVM estimates the hyperplane that separates the two classes in the feature space without modeling the distribution of these classes.

Some proposed methods also take an indirect approach to the binary classification task, such as (Resch et al., 2018) where Latent Dirichlet Allocation (LDA) (Blei et al., 2003) is used for topic modeling, and the resulting topic clusters are then analyzed further.

### 4.3.2 Neural networks

In recent years, neural networks have come to the forefront of research. In contrast to the models in the previous section, deep neural networks allow for more powerful and complex modeling, but also require more data and computational resources to train them, and their decisions are often less transparent. The last point can be particularly grave if critical decisions are made based on these models. Another difference is that they commonly do not use linguistically motivated features as their inputs, but instead use word or sentence embedding layers at the inputs, which are neural networks themselves. These embeddings are often pre-trained on even larger data sets, but can also be integrated into the training process for finetuning or training from scratch.

**Neural network features & embeddings**

As mentioned, hand-crafted features have become more and more replaced with automatically trained word embeddings since their inception in 2011 (Collobert et al., 2011). These embeddings are neural networks themselves, and are part of the complete classification network. Multiple refinements have been proposed over the years. Many approaches for crisis tweet detection employ *Word2vec*, a pre-trained word embedding that was first presented in 2013 (Mikolov et al., 2013) and has since been expanded in various ways. A version specifically trained on crisis tweets is presented in (Imran et al., 2016b). Burel et al. (2017a) integrate semantic concepts and entities from *DBPedia*[17]. *GloVe* (Pennington et al., 2014) and *fastText* (Joulin et al., 2016) embeddings follow a similar idea, and are expanded for multilingual tweet classification in (Lorini et al., 2019) and (de Bruijn et al., 2020) respectively, based on the adaptation method proposed by Lample et al. (2018).

In the past two years, BERT (Devlin et al., 2019) embeddings and their various offshoots have become very popular (McCreadie et al., 2020). These embeddings still function on the word level, but take complex contexts into account. A crisis-specific version is proposed in (Liu et al., 2020). In another direction, embeddings that do not represent words but whole sentences are also becoming used more widely, e.g. in (Kruspe, 2020; Kruspe et al., 2020; Wiegmann et al., 2020b). The most prominent example is the Universal Sentence Encoder (USE) (Cer et al., 2018) and its multilingual version (MUSE) (Yang et al., 2019).

In most cases, versions of embeddings that are pre-trained on large text corpora are used. These corpora are not necessarily social media texts or crisis-related, but the models have been shown to produce good results anyway. The advantage of using pre-trained models is that they are easy to apply, and do not require as much training data (Wiegmann et al., 2020b). In the case of sentence-level embeddings, their usage also leads to a simplification of the subsequent network layers as the embeddings

---

[17]https://wiki.dbpedia.org/

themselves already capture the context of the whole sentence. As mentioned above, versions finetuned to the task are also available for many common embeddings. A comparison of various word and sentence embeddings for crisis tweet classification can be found in (ALRashdi and O'Keefe, 2019).

It should also be mentioned that occasionally, deep models also utilize the linguistic features described above, e.g. (Ning et al., 2019). In the first iteration of the TREC-IS challenge, several approaches produced good results with such hand-crafted features as well (McCreadie et al., 2019). Their advantage lies in the fact that they do not need to be trained, and can therefore work with a small amount of data, which may sometimes be the case in new crises.

**Classification networks**

Extracted features, which may be embeddings are then fed into a subsequent neural network. In most crisis-related use cases, these will be classification models, although regression models are occasionally used for binary concepts like relevance, priority, or similarity, as well as sentiment. Commonly, text processing tasks employ Recurrent Neural Networks to leverage longer context, but in short text tasks, Convolutional Neural Networks (CNNs) are more popular.

Caragea et al. (2016) first employed CNNs for the classification of tweets into those informative with regards to flood events and those not informative. Lin et al. (2016) also applied CNNs to social media messages, but for the *Weibo* platform instead of Twitter. In many of the following approaches, a type of CNN developed by Kim for text classification is used (Kim, 2014), such as in (Burel and Alani, 2018; de Bruijn et al., 2020; Kersten et al., 2019). A schematic is shown in figure 2. These methods achieve accuracies of around $80\%$ for the classification into related and unrelated tweets. In (Burel and Alani, 2018) as well as in (Burel et al., 2017a) and (Nguyen et al., 2016b), this kind of model is also used for information type classification.

Recently, these CNN architectures have been expanded in different directions. Ning et al. (2019) show a multi-task variant. In (Burel et al., 2017a), a CNN with word embedding inputs is combined with one for semantic document representations. The resulting system is packaged as *CREES* (Burel and Alani, 2018), a service that can be integrated into other platforms similar to *AIDR*. Snyder et al. (2020) and Nguyen et al. (2016b) show active learning approaches that allow adapting the CNN over the progress of a crisis as new tweets arrive, dovetailing with the crowdsourcing systems described above. More novel approaches for adaptation to actionability are described in the next section.

## 4.4 Adaptation to actionability

All of the approaches mentioned above aim to generalize to any kind of event on tweet level without any *a priori* information, and can therefore not easily adapt to specific use cases. The transferability of pre-trained models to new events and event types is thoroughly investigated in (Wiegmann et al., 2020b). A real-world system may not need to be restricted in this way; in many cases, its users will already have some information about the event, and may already have spotted tweets of the required type. This removes the need to anticipate any type of event. It also directs the system towards a specific event rather than any event happening at that time.

As a consequence, a shift from static pre-trained models to more adaptable and flexible machine learning methods is re-

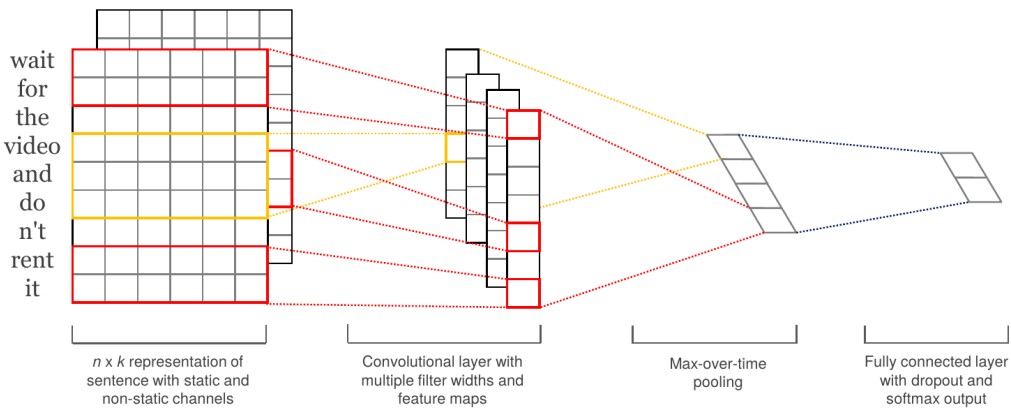

**Figure 2.** CNN for text classification as proposed by Kim (2014).

quired. Approaches such as semi-supervised learning of regression model ensembles (Kejriwal and Zhou, 2019), domain adaptation (Mazloom et al., 2019; Poblete et al., 2018), as well as active, incremental and online learning using Random

Forests (Kaufhold et al., 2020) demonstrate that traditional pre-trained models can also be utilized in a more interactive fashion and therefore have the potential to better fit to needs of emergency responders. With respect to deep learning, Li et al. (2018) and Mazloom et al. (2019) show that models adapted to the domain of the event can perform better than generalized models. Alam et al. (2018a) propose an interesting variant for neural networks: Their system includes an adversarial component which can be used to adapt a model trained on a specific event to a new one (i.e. a new domain). Pre-trained embeddings play a key

role in transfer learning or finetuning to new events, as they provide a large amount of pre-existing linguistic knowledge to the model, and therefore reduce the necessity for large amounts of training data (Snyder et al., 2020; Wiegmann et al., 2020b). In addition to their usage as classification inputs, embeddings can also be used in other ways, such as key- or descriptive word expansion (Viegas et al., 2019; Qiang et al., 2019), clustering (Hadifar et al., 2019; Comito et al., 2019), queries, or summarization (Singh and Shashi, 2019).

Kruspe et al. (2019) propose a system that does not assume an explicit notion of relatedness vs. unrelatedness (or relevance vs. irrelevance) to a crisis event. As described above, these qualities are not easy to define, and might vary for different users or different types of events. The presented method is able to determine whether a tweet belongs to a class (i.e. a crisis event or a desirable topic in a certain use case) implicitly defined by a small selection of example tweets by employing few-shot models. The approach is evaluated in more detail in (Kruspe, 2019).

In the broader picture of detecting actionable information, a trade-off between the flexibility of automatic data stream analysis methods and the available expertise and resources is required. Even though analysis on the tweet level may be fast and can be automated, this approach is quite restrictive because contextual information in terms of semantically similar message contents as well as developments over time and location are not taken into account. As a consequence, parallel events and discussions

are difficult to distinguish at this stage. We therefore propose to split the task of identifying actionable information with a spe-
cific thematic focus into two steps: (1) Data stream overload reduction with a general, potentially automated and pre-trained model for classifying disaster- or incident-related tweets, and (2) applying (one or even more subsequent) methods that allow for tailored contextual, semantic and/or interactive analyses of the filtered results. This type of approach has, for example, been investigated in (Alam et al., 2020; Kersten and Klan, 2020), and is intended to offer a modular and flexible set of well-understood methods addressing user-specific sub-tasks, and to provide insights on different granularity levels. Compared to an end-to-end ("black box") approach comprising multiple tasks, modularity helps to keep the complexity low for each sub-task. Furthermore, this worfklow supports process interpretability and offers the ability to transparently fuse, combine, or jointly interpret the results from each actionability sub-task.

Methods suitable for in-depth analyses of pre-filtered (i.e. crisis-related) tweets can be grouped into supervised, unsupervised, and hybrid ones. One straightforward approach is the tweet-wise classification into information classes described earlier. The aforementioned data sets *CrisisLexT26*, *CrisisMMD*, *TREC-IS 2019B*, and *SSM* provide example tweets for such classes or ontologies, which were defined in cooperation with emergency managers or agencies. As suggested in (McCreadie et al., 2020), a specific (but not necessarily fixed) subset of information classes can then be analyzed more closely as they represent actionable topics, like "Request-SearchAndRescue" or "Report-EmegingThreats". Additionally, a tweet-wise ranking according to a priority level (e.g. "low", "medium" and "high"), either through classification or through regression, is useful for information prioritization. Ranking tweets via deep learning-based and handcrafted features describing the quality of content (Ibtihel et al., 2019) in order to find fact-checkable messages (Barnwal et al., 2019) or informative content based on multi-modal analyses (Nalluru et al., 2019) are further promising options.

However, tweet-wise analyses alone do not exploit the full potential offered by the Twitter data stream. Important aspects, like aggregating messages, assessing the credibility or geolocation accuracy of a single message/information, and understanding the "big picture" of a situation can significantly be supported by integrating context. In particular, the utilization of unsupervised methods enables a flexible capturing of unforeseen events, discussions, developments, and situations that indicate the need for action.

Identifying significant increases of "bursty keywords" might be a first option for detecting events, like earthquakes (Poblete et al., 2018), but this approach alone tends to produce quite noisy results (Ramachandran and Ramasubramanian, 2018). Topic modeling techniques, like Non-negative Matrix Factorization (NMF) and Latent Dirichlet Allocation (LDA), are commonly used to identify discussed topics (e.g. (Casalino et al., 2018) and (Resch et al., 2018)). Furthermore, clustering techniques that utilize spatial (Ester et al., 1996), spatio-temporal (Birant and Kut, 2007; Lee et al., 2017), and content-based features (Mendonça et al., 2019; Comito et al., 2019; Singh and Shashi, 2019; Fedoryszak et al., 2019) as well as combinations of these (Nguyen and Shin, 2017; Zhang and Eick, 2019) are available. A quite interesting and effective approach lies in directly using word or sentence-embeddings to semantically cluster tweets for various tasks, like the detection of topics (de Miranda et al., 2020), events (Ertugrul et al., 2017), or novelty during crises (Kruspe, 2020). A further promising direction is the combination of pre-trained models and unsupervised methods like the aforementioned clustering. In (Bongard, 2020; Kersten et al., 2021), for example, an unsupervised grouping of incoming tweets helps to keep track of all discussed topics. A sim-

ple list of keywords or hashtags together with pre-trained models then support the automated identification of topic-specific, crisis-related, or actionable clusters. An exemplary result based on the *Events2012* data set is depicted in figure 3.

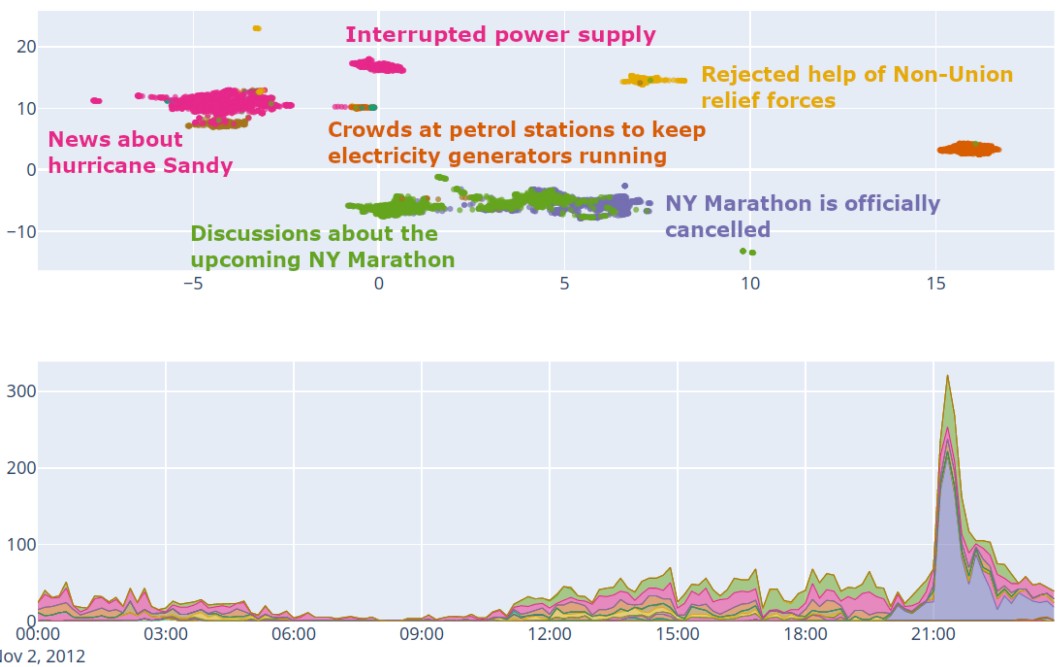

**Figure 3.** Top: 2D visualization of clusters containing the keyword "Long Island" identified on October 14, 2012 (arbitrary dimensions). Bottom: Tweet counts over time (GMT) per cluster. Source: Bongard (2020)

The methodological improvements mentioned above may still not be sufficient for real-world scenarios. Limited personal or computational resources and expert domain knowledge paired with time pressure and data uncertainty motivate the integration of machine learning methods into "systems" that allow to better interact, adjust, summarize, and visualize data analysis results. In this regard, McCreadie et al. (2016) propose an Emergency Analysis Identification and Management System (EAIMS) to

enable civil protection agencies to easily make use of social media. The system comprises a crawler, service, and user interface layer and enables real-time detection of emergency events, related information finding, and credibility analysis. Furthermore, machine learning is utilized over data gathered from past disasters to build effective models for identifying new events, tracking developments within those events, and analyzing those developments to enhance the decision-making processes of emergency response agencies. The recently proposed decision support system Event Tracker (Thomas et al., 2019) aims at providing a

unified view of an event, integrating information from news sources, emergency response officers, social media, and volunteers.

## 5 Challenges

None of the approaches presented are able to solve the problem of detecting tweets in disaster events perfectly. In some respects, this is due to technical limitations; however, there are several difficulties immanent to the task itself, which we will discuss in this section.

**Ambiguous problem definition** As stated throughout the paper, the task of tweet detection in disasters is ill-defined and heavily dependent on the use case. Annotation experiments also show that even if the goal is clearly stated, inter-rater agreement is commonly low, with raters often interpreting both the problem statement as well as tweet content very differently (Stowe et al., 2018). This problem becomes even more emphasized when annotating more fine-grained labels, e.g. for content type classes or for priority. Current research suggests a shift from the target of situational awareness to user-specific actionability.

**Linguistic difficulties and language variety** As mentioned above, most data sets and, accordingly, methods for automatic tweet detection focus on English-language data. This would often not be the best choice in a real-world scenario; multilingual methods are necessary.

Apart from the question of the language itself, Twitter users frequently utilize an highly idiosyncratic style of writing. Due to the character limitation, words are often abbreviated and grammar is compressed. In contrast to e.g. newspaper articles, user-generated content is relatively noisy, containing lots of erroneous or specialized spelling variations. Additionally, interpretation of tweet content frequently requires (cultural) context knowledge.

**Data limitations, legal and privacy issues** As mentioned above, Twitter is one of the few popular social media platforms providing an access API to its data to outside users. Despite this, however, limitations exist. For non-paying users, only 1% of the live data of each second can be collected automatically via Twitter's streaming API. For past events, the search API can be utilized, but this only returns tweets still in the search index, which is usually valid for around one week. Older tweets can be retrieved by their ID, but this does not allow for a flexible search. As a free user, the download rate is limited to 18,000 tweets per 15 minutes. Twitter also offers several paid options (called "firehoses") to access more live data, but these are somewhat intransparent. An in-depth analysis of the effect that these limitations can have on research is given in (Valkanas et al., 2014).

Twitter also forbids direct redistribution of tweet content, meaning that the described data sets are only available as lists of tweet IDs. This introduces two difficulties: One, retrieving the actual tweet content ("hydrating") can take a very long time for large data sets due to the rate limit. Two, tweets may become unavailable over time because their creator deleted them or their whole account, or because they were banned. In some cases of older data sets, this means that a significant portion of the corpus cannot be used anymore, impeding reproducibility and comparability of published research.

Apart from access limitations, Twitter and legal restrictions also regulate what researchers are allowed to do with this data. As an example, the Twitter user agreement states (Twitter, Inc., 2020):

"Unless explicitly approved otherwise by Twitter in writing, you may not use, or knowingly display, distribute, or otherwise make Twitter Content, or information derived from Twitter Content, available to any entity for the purpose of: (a) conducting or providing surveillance or gathering intelligence, including but not limited to investigating or tracking Twitter users or Twitter Content; (b) conducting or providing analysis or research for any unlawful or discriminatory purpose, or in a manner that would be inconsistent with Twitter users' reasonable expectations of privacy; (c) monitoring sensitive events (including but not limited to protests, rallies, or community organizing meetings); or (d) targeting, segmenting, or profiling individuals based on sensitive personal information, including their health (e.g., pregnancy), negative financial status or condition, political affiliation or beliefs, racial or ethnic origin, religious or philosophical affiliation or beliefs, sex life or sexual orientation, trade union membership, Twitter Content relating to any alleged or actual commission of a crime, or any other sensitive categories of personal information prohibited by law."

Many interesting research questions are not identical, but related to problematic usages described in this statement, e.g. inference on a user basis or monitoring of protests. Researchers must therefore be careful not to step into prohibited territory.

**Lack of geolocation**  In a disaster context, knowing exactly where a tweet was sent is often crucial to the usability of this information. Twitter provides several ways of detecting geolocation. The most precise of them is the option for users to send their coordinates along with the tweet. However, only about 1% of tweets contain this information (Sloan et al., 2013). A tweet's location can also be estimated from the location stated in the user profile, or by analyzing the tweet's content with regards to mention of geolocation. For operationalization, a geocoding to coordinates is then required, which can be provided by services such as Google Maps or OpenStreetMap's Nominatim. Unfortunately, these geolocations are prone to errors, e.g. because a user mentions a position other than their own, because they might be traveling, or because the center coordinates of a city are to imprecise to be usable. Geocoding, i.e. the prediction of tweet locations from other sources such as the text content, is also an active area of research (e.g. (Qazi et al., 2020; Brouwer et al., 2017)).

## 6   Related tasks

Once tweets related to a disaster event have been discovered, many further analysis steps are possible. We will only touch upon those briefly here. As described in section 3, some of the available data sets have already been annotated with these additional concepts.

A popular next step that many automatic approaches already include is the classification into semantic or information type classes. Such classes may include sentiments, affected people seeking various types of assistance, media reports, warnings and advice etc. No common set of such classes exists; in the *CrisisNLP* and *CrisisLexT26* corpora, 9 and 7 classes are used respectively with some overlap. For the TREC Incident Streams challenge, potential end users were questioned about their

classes of interests, resulting in a two-tier ontology with 25 classes on the lower tier. As an added difficulty, classes often overlap in tweets; for these reasons, TREC allows multiple labels per tweet. Furthermore, annotators often disagree whether an information type is present in a tweet.

Another way of further discerning between tweets is a distinction between levels of informativeness or priority. This can be implemented either with discrete classes (low/medium/high importance), on a continuous numerical scale, or as a ranking of

tweets. The *CrisisLexT26* and *TREC-IS 2019A* data sets contain such annotations.

Apart from approaches processing single tweets, the analysis of the spatio-temporal distribution and development of discussed topics within affected areas at different scales may provide valuable insights (Kersten and Klan, 2020). Other research focuses on the detection of specific events, or types of events (e.g. floods, wildfires, or man-made disasters) (e.g. Burel et al. (2017b)). This can often be helpful when social media is used as an alerting system. Additionally, models specialized to event types

can be more precise and allow for different distinctions than general-purpose models (Kersten et al., 2019; Wiegmann et al., 2020b); detection of the event type enables the automatic selection of such a more specialized method.

Apart from these text-based tasks, image analysis can also be a helpful source of information. As an example, images posted on social media can be used to determine the degree of destruction in the aftermath of a disaster (Alam et al., 2017; Nguyen et al., 2017b).

As suggested in section 4, taking a larger variety of semantic concepts into account could lead to a possible solution of the problem of automatic actionability detection. These concepts can be combined in intelligent and adaptable ways to zone in on what exactly are relevant tweets to a user.

## 7    Future work

Many very interesting new analysis tasks are thinkable based on the detection methods described so far, particularly when

employing automatic methods. A good starting point to identify relevant practical issues related to acquisition tasks that could potentially be solved by analyzing social media data is provided in (Wiegmann et al., 2020c). Here, opportunities and risks of disaster data from social media are investigated by means of a systematic review of currently available incident information.

One aspect that has not been considered in research so far is how an event changes over time. New approaches could be used to analyze the spatiotemporal development of disasters, and how this could be utilized in disaster prevention. During the course

of an event, clustering methods could be employed to rapidly detect novel developments such as sub-events or new topics. This is particularly relevant for relief providers, who require extremely fast situation monitoring.

As described in section 5, localizing information coming from Twitter is often a challenge. Approaches that are able to deal with this lack of information are necessary. This could be implemented either by deriving location by some other means, or by spatiotemporal and semantic analysis of large sets of tweets to cross-reference and check information.

As mentioned above, languages other than English have also usually not been included in research on this topic. Multilingual approaches would be a very helpful next step to facilitate usage of such methods in regions of the world where English is not the main language. Another aspect of the data that has not been used often so far are images posted by users. In particular, a

multimodal joint analysis of text and images is very interesting from both the research as well as the usage perspective. The *CrisisMMD* data set is an interesting first step in this direction.

As described in section 4, some crowdsourcing approaches already integrate machine learning-based methods. In future work, expanding human-in-the-loop approaches would be very useful.

In general, social media is usually not the only source of information and cannot provide a full picture of the situation. Therefore, an integration with other information sources, such as earth observation data, media information, or governmental data, is highly relevant. de Bruijn et al. (2020) present a first foray by combining social media information with hydrological data.

As described, a large step towards making automatic tweet detection approaches more useful in real-life systems lies in their adaptability to the desired use cases. We have identified three promising research directions in this paper:

1. Exploiting various concepts and other analysis methods suggested in this section to allow users to flexibly define actionability and detect tweets based on this definition.

2. Machine learning models that can adapt to new use cases, e.g. through active learning or few-shot modeling with the
involvement of users, through domain adaptation, or through novelty detection.

3. Complex systems that integrate automatic tweet analysis with available expertise and other resources, e.g. by combining an automatic pre-filtering step with dedicated methods for a specific actionability scenario (either manual or also automatic).

## 8   Conclusions

In this review paper, we gave an overview over current methods to detect tweets pertaining to disaster events. As a major hindrance, we identified the necessity for an exact definition of the desired tweets. Conventionally, automatic recognition of tweets aims to achieve a generalized situational awareness, utilizing the ill-defined concepts of "relatedness", "informativeness", or "relevance". In real-world scenarios, however, the question which tweets should be detected depends on the use case, and has been framed as the concept of actionability in recent research. Most data sets and applications do not yet offer this flexibility.

We compare various crisis tweet data sets available online. Unfortunately, these usually only provide ID's of the tweets, which leads to changes in the data sets over time. In addition, labels are usually only provided for the described static binary concepts (related/informative/relevant), and definitions do not match across data sets. Nevertheless, these collections are a very useful basis for analyzing user behavior and for developing new models. They also frequently offer annotations for other concepts, such as information types or sources. We believe integrating these concepts in future approaches could lead to more flexibility
in the domain of actionability.

On the methodical side, there are three main ways to approach the problem: Filtering tweets by characteristics such as location and contained keywords or hashtags, crowdsourcing, and machine-learning based methods. Each of these has its advantages and disadvantages, but machine learning appears to be the current main avenue of research with big improvements in the past few years. Once again, most methods from the past few years follow a static ontology, but there is now a development towards

novel approaches that allow for flexible adaptation to user-based actionability definitions, e.g. via few-shot learning based on a small number of example tweets or by detecting specific topical clusters of tweets.

Besides the definition problem, other difficulties include the subjectivity of classes and tweet interpretations, data limitations, linguistic difficulties, and legal issues. Nevertheless, large strides have been made in the past years to tackle this problem, and research in this area remains highly active. Many related and novel analysis tasks are possible in the future. To mention a specific example, the COVID-19 pandemic has already led to a number of novel data sets and approaches. It will be interesting to see how these develop further for such a long-term crisis. For further reading on the topic of crisis informatics as a whole, we recommend the bibliography provided in (Palen et al., 2020).

*Author contributions.* Anna Kruspe wrote this paper with assistance and input from Jens Kersten and Friederike Klan. Jens Kersten contributed ideas and text to sections 2, 3, 4.3, and 4.4.

*Competing interests.* The authors declare that they have no conflict of interest.

*Disclaimer.* TEXT

*Acknowledgements.* TEXT

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
