# Peer review of "Review article: Detection of actionable tweets in crisis events"

_Natural Hazards and Earth System Sciences, 2020_

## Referee Comment (RC1) · Anonymous Referee #1 · 11 Aug 2020

General Comments This is a review paper summarizing at a high level currently available datasets and approaches that tackle crisis analytics on Twitter data, as well as highlights some challenges. The paper is overall well written and should be understandable by an in-expert audience. It may be valuable as an entry point for new researchers looking to work in the area of crisis informatics (although there are already good resources available in this area (e.g. the Big Crisis Data book). On the other hand, the paper is quite shallow in terms of detail in all aspects and so only acts as a guidepost for further reading on the subject. There are no individual experiments by the authors provided in the paper.

Detailed Comments The core limitation of this work from my reading is that it tries to summarize too many areas of the field of crisis informatics and so currently does not

provide enough detail on any one to provide significant insights that add value over the individual works. For example, Section 2 provides a brief summary of some of the definitions that different groups have used for analysing crisis content, but don't go into detail on what works are compatible with one another, or indeed provide information on the definition of 'Informative' regards to who is used in each work. As a result it is not clear to someone just entering the field what they should read. Similarly, Section 3 highlights some datasets used in crisis informatics, but Table 2 only lists tweet counts, not the volume labelled and what was labelled or for what task. Which datasets are complementary? Which are easy to work with? What datasets do the authors recommend researchers use?

The second question I have for this overview is whether COVID-19 datasets belong in this study or whether they should be considered separately. COVID is quite a different task compared to natural or manmade disasters, as there typically is not a strong timeliness component to related information needs. Hence, the definition of what is informative for pandemics and the associated target user groups are very different. I would recommend at least adding some discussion in Section 2 on this point.

Third, I would recommend structuring the discussion on the machine learning aspects along the lines of what task is being investigated, inputs, features and models. Indeed, it would be valuable to get some idea of how many works use each different approach, as well as get some data on the prevalence of different feature and text representation approaches used in the different works and critically, what patterns emerge on what works.

Other notes: - Table 2 should highlight the differences between labelled and un-labelled tweets - I believe the statistics for the TREC-IS data in particular is out-of-date, see the ISCRAM 2020 paper http://trecis.org/2020/ISCRAM_2020_TREC_IS.pdf

---

## Referee Comment (RC2) · Anonymous Referee #2 · 12 Sep 2020

General comments: The review paper summarizes research on the automatic process-ing of social media messages in crisis situations. The paper outlines varying concepts of information, relevance, available datasets, filtering approaches and associated chal-lenges. Overall, and echoing the concern of the previous reviewer, the paper can serve as an introduction to this research but lacks the comprehensive detail or specific focus that would recommend it to experts. The authors might consider approaching the re-view from the perspective of one of the challenges highlighted in section five with the goal of providing an overview of approaches addressing this challenge and revealing new directions for research.

Specific comments: The different understandings of information relevance between research focusing on technical challenges related to processing social media messages and research focusing on the information requirements of end users, i.e., emergency responders, deserves further attention. As the authors point out, the former includes different concepts of information relevance—"related", "relevant", and "informative"—that are not often compared or resolved in research addressing technical challenges. Moreover, compared to research on end user requirements, these definitions are considered much too coarse grained. When employed in the design of social media filtering tools, these designs may provide information that is somehow related to a crisis but typically unrelated to responders' information needs which not only vary by role and crisis, but by dynamic contextual factors such as the availability of information from existing, traditional information sources. Recent research has considered these challenges by turning to the concept of "actionability" to describe information relevance from the end user perspective of emergency responders (Kropczynski et al., 2018; McCreadie e a., 2020; Zade et al., 2018).

As the paper provides a cursory overview of available datasets, filtering approaches and associated challenges, greater detail might be included in each. As described above, selecting a particular research challenge as a way to focus the review and discuss in greater detail the contributions of these studies with respect to this challenge would offer readers more insight into the research space.

There are additional datasets that might be included, such as listed in Palen et al. (2020) and Grace (2020).

McCreadie, R., Buntain, C, Soboro, I. (2020). Incident Streams 2019: Actionable Insights and How to Find Them. In Proceedings of the 17th ISCRAM Conference (pp. 77-760).

Kropczynski, J., Grace, R., Coche, J., Jalse, S., Obeysekare, E., Montarnal, A., ... & Tapia, A. (2018). Identifying actionable information on social media for emergency dispatch. In Proceedings of the ISCRAM Asia Pacific (pp. 1-11).

Zade, H., Shah, K., Rangarajan, V., Kshirsagar, P., Imran, M., & Starbird, K. (2018).

From situational awareness to actionability: Towards improving the utility of social media data for crisis response. Proceedings of the ACM on human-computer interaction, 2(CSCW), 1-18.

Palen, L., Anderson, J., Bica, M., Castillos, C., Crowley, J., Díaz, P., ... & Kogan, M. (2020). Crisis Informatics: Human-Centered Research on Tech & Crises. Retrieved from https://hal.archives-ouvertes.fr/hal-02781763/document

Grace, R. (2020). Crisis social media data labeled for storm-related information and toponym usage. Data in brief, 105595. doi:10.1016/j.dib.2020.105595

---

## Author Comment (AC1) · 8 Oct 2020

Dear reviewer,

thank you very much for your comments. The main issue you mention is that the review paper is somewhat unfocused and therefore does not provide enough guidance for someone just entering the field. We certainly understand this criticism. In the next revision of the paper, we will add more of our own experience to provide recommendations and disentangle some of the questions that new researchers may have. (Reviewer 2 suggests using one of the challenges from section 5 as a starting point here, which we think is a good idea).

Thank you also for your comment on the inclusion of COVID-19 data. We deliberated

for a while on whether to include this or not and then decided for it due to the current urgency for analysis of such data. If there is a consensus that it would be better to drop this topic from the paper, we would be very open to perhaps moving this into a follow-up paper instead, especially once there is more work on COVID-19.

To your third point: The machine learning approaches are, in our opinion, the focal point of the review paper. We would therefore be very willing to expand this section and provide more context. We think this also correlates with the first point by providing a more comprehensive state of the art to new researchers.

To summarize, we will focus the paper more on certain topics and add our own perspective for researchers new to the field in the next revision. Depending on the opinions of the other reviewers, we will either remove the COVID-19 data sets or add some discussion about their particular challenges.

---

## Author Comment (AC2) · 8 Oct 2020

Dear reviewer,

thank you for your comments. As you mention, your main criticism overlaps with that of reviewer 1: The paper could be more focused and offer more insights to new researchers in the field. We think that your idea to use one of the challenges from section 5 as a starting point is a very good one, and will do this in the next revision.

Thank you also for the additional references and the recommendation to extend the discussion of relatedness/relevancy/informativeness along the lines of actionability. This will add a nice resolution to section 2. We will also include the additional data sets you mentioned.

[Figure]

To summarize: In the next revision, we will focus the paper more (on one of the section 5 challenges) and add more of our own experiences to provide guidance to new researchers.
* * *

---

## Author Response (AR1)

**Review 1**

**General Comments** This is a review paper summarizing at a high level currently available datasets and approaches that tackle crisis analytics on Twitter data, as well as highlights some challenges. The paper is overall well written and should be understandable by an in-expert audience. It may be valuable as an entry point for new researchers looking to work in the area of crisis informatics (although there are already good resources available in this area (e.g. the Big Crisis Data book). On the other hand, the paper is quite shallow in terms of detail in all aspects and so only acts as a guidepost for further reading on the subject. There are no individual experiments by the authors provided in the paper.

**Detailed comments** The core limitation of this work from my reading is that it tries to summarize too many areas of the field of crisis informatics and so currently does not provide enough detail on any one to provide significant insights that add value over the individual works. For example, Section 2 provides a brief summary of some of the definitions that different groups have used for analysing crisis content, but don't go into detail on what works are compatible with one another, or indeed provide information
on the definition of 'Informative' regards to who is used in each work. As a result it is not clear to someone just entering the field what they should read.

➔ We have expanded this section to give recommendations for handling this issue in future research. Newer publications have shifted to a more user-centric definition (actionability) and we show this as a path forward both in this section and in the rest of the paper.

Similarly, Section 3 highlights some datasets used in crisis informatics, but Table 2 only lists tweet counts, not the volume labelled and what was labelled or for what task. Which datasets are complementary? Which are easy to work with? What datasets do the authors recommend researchers use?

➔ Labeling information has been added to the table. At the end of the section, we have added a comparison of what data sets are useful for what purposes.

The second question I have for this overview is whether COVID-19 datasets belong in this study or whether they should be considered separately. COVID is quite a different task compared to natural or manmade disasters, as there typically is not a strong timeliness component to related information needs. Hence, the definition of what is informative for pandemics and the associated target user groups are very different. I would recommend at least adding some discussion in Section 2 on this point.

➔ COVID-19 data sets have been removed from the overview, and COVID-19 data analysis is now mentioned as an interesting new research task. We think it would be interesting to get started on a separate review paper about COVID-19 data sets and approaches in a year or so.

Third, I would recommend structuring the discussion on the machine learning aspects along the lines of what task is being investigated, inputs, features and models. Indeed, it would be valuable to get some idea of how many works use each different approach, as well as get some data on the prevalence of different feature and text representation approaches used in the different works and critically, what patterns emerge on what works.

➔ Section has been expanded with more ML approaches (particularly non-neural network ones) and a table comparing methods, features, tasks, and used data. Subsections have been

restructured to better compare used feature and ML methods.

Other notes: - Table 2 should highlight the differences between labelled and un-labelled tweets - I believe the statistics for the TREC-IS data in particular is out-of-date, see the ISCRAM 2020 paper http://trecis.org/2020/ISCRAM_2020_TREC_IS.pdf

➔ Has been updated, table now includes statistics of labeled vs. unlabeled tweets

**Review 2**

**General comments** The review paper summarizes research on the automatic processing of social media messages in crisis situations. The paper outlines varying concepts of information, relevance, available datasets, filtering approaches and associated challenges. Overall, and echoing the concern of the previous reviewer, the paper can serve as an introduction to this research but lacks the comprehensive detail or specific focus that would recommend it to experts. The authors might consider approaching the review from the perspective of one of the challenges highlighted in section five with the goal of providing an overview of approaches addressing this challenge and revealing new directions for research.

➔ We have chosen the challenge of varying definitions of relatedness/relevance/importance as this is also the start of the paper. The paper now provides possible solutions in each section.

**Specific comments** The different understandings of information relevance between research focusing on technical challenges related to processing social media messages and research focusing on the information requirements of end users, i.e., emergency responders, deserves further attention. As the authors point out, the former includes different concepts of information relevance ("related", "relevant", and "informative") that are not often compared or resolved in research addressing technical challenges. Moreover, compared to research on end user requirements, these definitions are considered much too coarse grained. When employed in the design of social media filtering tools, these designs may provide information that is somehow related to a crisis but typically unrelated to responders' information needs which not only vary by role and crisis, but by dynamic contextual factors such as the availability of information from existing, traditional information sources. Recent research has considered these challenges by turning to the concept of "actionability" to describe information relevance from the end user perspective of emergency responders (Kropczynski et al., 2018; McCreadie et a., 2020; Zade et al., 2018).

➔ Has been included and is now a focal point of the paper as described above.

As the paper provides a cursory overview of available datasets, filtering approaches and associated challenges, greater detail might be included in each.

➔ Data and approach sections have been expanded and the comparison has been made more detailed

As described above, selecting a particular research challenge as a way to focus the review and discuss in greater detail the contributions of these studies with respect to this challenge would offer readers more insight into the research space.
There are additional datasets that might be included, such as listed in Palen et al.
(2020) and Grace (2020).

➔ Grace data set and additional literature have been included (including Palen et al.)

---

## Author Response (AR2)

**Review 1**

The paper describes how information extracted from the micro-blogging platform Twitter can serve Crisis Management during a disaster rather than finding actionable tweets.

The structure of the paper is clear, and the data are exhaustively detailed. Therefore the presentation quality is good. I have significant doubts about the scientific significance and the quality of the outcomes.

**Scientific significance**

Contribution to the research community:

a-The vast majority of SM research in DRM uses Tweets as they are the 'de-facto' only open and free source of SM data

b-The authors start the paper in the quest for actionability but then end up describing mostly research works used for classifying relevance of tweets (4.2 and 4.3). Neither the datasets described classified tweets as 'actionable' but rather as informative or relevant.

➔ This difficulty is pointed out in the paper; previous research mainly focuses on "relevance" or "informativeness", not on actionability. We have now clarified that these works form a basis for adaptation to actionability.

As it is now, the paper offers an update of similar works already published (HAL, Reuter work to cite two). The novel information is the in-depth description of the models and technologies used for automated classification of tweets. The authors should leverage on the richness of the examples offered to better present the state of the art of several lines of research according to crisis management needs.

**Scientific quality**

My concern is that the paper's goal is not clear (or the research question not posed correctly?). I cannot understand if the article wants to offer a path for future research on finding serviceable information or describing the state of the art of SM for Crisis Management.

In the first case, the paper misses a solid base of works for supporting a choice; in the second case, there is an unclear definition for 'actionable information'.

➔ In a sense, both. We present the current state of the art, but point out the unclear definition of what exactly approaches published so far are supposed to classify. As suggested by the other reviewers, we then introduce the concept of actionability in the sense of letting the user define what is relevant to them/their use case. On methods, we discuss first what is already there, and second how actionability can be introduced. The paper has been updated to clarify our approach to the problem and the progression from existing works to methods for actionability. The definition of actionability is also stated more clearly now.

A tentative definition is quickly described in the framing of the problem, but then it is not followed in the next chapters of the article (4.2 and 4.3). The authors describe models for classification for situational awareness in the machine learning approaches justifying it as possible pre-selection for further analysis for finding actionable information. That's perhaps confusing. There should be a more

precise definition of classes of actions/decisions and clear links between each data/algorithm/technology described and such classes.

➔ We explicitly do not focus on specific use cases or events because we are more interested in making systems usable for various ones, even those that may come up in the future and are not yet predictable (as we are now seeing in the pandemic). We have now included this motivation in the paper.

Another concern is that if the paper aims to offer a starting point for further reading, I see the risk of highlighting only some branches of the research from the community of researchers studying the intersection of Crisis Informatics and Social Media, which is significant and growing (focus seems only on the authors that published data in the datasets section/par).

Some missing ref for instance (easily found on Arxiv or Google Scholar):

Purohit H. efforts for serviceability of information is missing (Purohit, Peterson).

H2020 funded Projects such as E2MC or I-REACT are not cited while they could be listed in the crowdsourcing approach (both produced publications).

Castillo is cited but not for his last works where Convnet is used for multilingual annotation (Lorini et al., ISCRAM 19 paper award)

Again no mention of De Bruijn or Brouwer works about the detection of impact (damage assessment) very relevant together with the cited Albuquerque research about mixing authoritative and social media data.

Poblete research in Disaster Management focuses on impact detection and multilingual text.

➔ References have been included

I also think leaving out the analysis of research on Images (growing area of study) from Social Media is also a missed opportunity to describe a growing source of informative data.

➔ We agree that this is a very interesting new development, but have not included it in the paper in order to maintain a text-centric focus. If there is interest in this, we are open to expanding the paper in this direction.

**Suggestion for revision:**

As it is now the work has a good depth, but I am afraid is mixing different researches and experiments. It would be better, in my opinion, to change the aim of the paper as to describe how Crisis Informatics research try to extract from SM information that can be 'fit-for-purpose' in the context of Crisis Management and then divide the work into purposes (finding requests for action, situational awareness, Impact assessment, ...).

Restructuring the paper according to 'purposes' or 'needs' would simplify the 'labelling' of the described works and streamline the narrative (extending the references of course).

➔ See above; we would prefer not to limit ourselves to certain use cases. We hope the new version of the paper conveys more clearly what we mean by focusing on actionability, and how this can be a path forward for many new use cases (i.e. the mentioned 'fit-for-purpose' data retrieval) in the future.
We thank the reviewer for their helpful comments and suggested references!

---

## Author Response (AR3)

**Author response**

We have now removed the sentences about COVID-19.

Once again, we would like to thank the reviewers and editor for their valuable comments.